# Assembly of Synaptic Protein–DNA Complexes: Critical Role of Non-Specific Interactions

**DOI:** 10.3390/ijms24129800

**Published:** 2023-06-06

**Authors:** Sridhar Vemulapalli, Mohtadin Hashemi, Anatoly B. Kolomeisky, Yuri L. Lyubchenko

**Affiliations:** 1Department of Pharmaceutical Sciences, University of Nebraska Medical Center, 986025 Nebraska Medical Center, Omaha, NE 68198-6025, USA; sridhar.vemulapalli@unmc.edu (S.V.); mohtadin.hashemi@unmc.edu (M.H.); 2Department of Chemistry-MS60, Rice University, 6100 Main Street, Houston, TX 77005-1892, USA

**Keywords:** site-search, threading, site-bound transfer, entropic effect, pre-synaptic, transient, synaptic protein–DNA complex, single-molecule-TIRF

## Abstract

The synaptic protein–DNA complexes, formed by specialized proteins that bridge two or more distant sites on DNA, are critically involved in various genetic processes. However, the molecular mechanism by which the protein searches for these sites and how it brings them together is not well understood. Our previous studies directly visualized search pathways used by SfiI, and we identified two pathways, DNA threading and site-bound transfer pathways, specific to the site-search process for synaptic DNA–protein systems. To investigate the molecular mechanism behind these site-search pathways, we assembled complexes of SfiI with various DNA substrates corresponding to different transient states and measured their stability using a single-molecule fluorescence approach. These assemblies corresponded to specific–specific (synaptic), non-specific–non-specific (non-specific), and specific–non-specific (pre-synaptic) SfiI–DNA states. Unexpectedly, an elevated stability in pre-synaptic complexes assembled with specific and non-specific DNA substrates was found. To explain these surprising observations, a theoretical approach that describes the assembly of these complexes and compares the predictions with the experiment was developed. The theory explains this effect by utilizing entropic arguments, according to which, after the partial dissociation, the non-specific DNA template has multiple possibilities of rebinding, effectively increasing the stability. Such difference in the stabilities of SfiI complexes with specific and non-specific DNA explains the utilization of threading and site-bound transfer pathways in the search process of synaptic protein–DNA complexes discovered in the time-lapse AFM experiments.

## 1. Introduction

The interaction between distant sites on DNA, mediated by specialized proteins, results in the formation of site-specific synaptic complexes (i.e., synaptosome) [1]. The formation of synaptic complexes is a fundamental step in several genetic processes, including gene regulation (e.g., the GalR repressor) [2,3], site-specific recombination (e.g., the Flp recombinases) [4], and various eukaryotic gene rearrangement systems, such as transposons [5], the Variable Diversity Joining (V(D)J) recombination system involving tetrameric RAG1/2 assembly [6], and HIV integration systems [7,8]. Despite the importance of the formation of synaptic complexes, the underlying mechanism of their formation is poorly understood [1,4,9,10]. A seminal theoretical model by Berg, von Hippel, and Winter (i.e., the BHW model) [11,12,13,14,15] has previously described the single-site-search process, where the proteins use sliding, jumping, and intersegmental transfer pathways in search for the specific site on DNA. However, the model is limited to the site-search process for a single site and is not applicable to proteins interacting with multiple specific sites on DNA. To understand these long-range interactions on DNA, we have previously used the restriction enzyme SfiI as a model system, which cleaves DNA after binding to two cognate sites using Mg^2+^ cations as cofactors [10,16,17,18,19,20,21,22,23].

Assembly of a synaptic complex with the DNA molecules containing two cognate sites results in looped DNA complexes with the loop sizes corresponding to the distances between the two cognate sites [9,10,19]. In our recent studies, we used time-lapse HS-AFM to directly visualize the dynamics of the site-search process by SfiI [24]. Using this approach, we visualized sliding and jumping pathways leading to the SfiI binding to DNA. The assembly of synaptosomes occurs via two new pathways. In one of them, SfiI is bound to a specific site and binds to a non-specific site forming a looped complex and threads DNA while searching for another specific site. Of note, the threading mechanism was previously visualized in the formation of synaptosome by the EcoRII restriction enzyme, [25,26] suggesting that this is a common site-search pathway for the assembly of synaptic protein–DNA complexes. In another pathway, the complex of SfiI bound to the specific sites jumps over a large distance, forming a looped complex. These studies led to the proposed site-search model for SfiI [24]. According to this model, the search for the first specific site can occur via the BHW model, followed by the formation of synaptic complexes via the new pathways described above. However, molecular mechanisms explaining these site-search processes remain unclear. 

In this study, we employed single-molecule total internal reflection fluorescence (TIRF) microscopy in the TAPIN (Tethered Approach for Probing Intermolecular Interactions) mode [27,28,29,30] to characterize the lifetime different SfiI–DNA complexes corresponding to the transient states of the synaptosome assembly. A theoretical model for the synaptosome assembly was built to explain the experimental observations. Specifically, the theory explains the elevated stability of pre-synaptic SfiI–DNA complexes by the increased dynamics of the non-specific duplex within the synaptosome. This finding provides a molecular mechanism for threading and site-bound transfer pathways in the site-search process for synaptic protein–DNA systems. 

## 2. Results

To characterize the lifetime of SfiI–DNA complexes, we used a tethered approach for probing intermolecular interaction (TAPIN). This method was developed and previously used to study the intermolecular interactions between biomolecules [27,28,29,30]. In this approach, a DNA duplex is covalently attached to the surface via a thiol moiety, and a fluorophore-labeled DNA duplex is added to the chamber, together with SfiI, and acts as the probe, as shown in Figure 1A. When no complexes between SfiI and the two DNA duplexes occur, no change in the fluorescence intensity is observed, as shown in Figure 1B. However, when there is an interaction between the SfiI and two DNA duplexes, a burst of fluorescence is detected, as shown in Figure 1C. The time between the sudden increase and the abrupt drop in the fluorescence intensity was considered the complex’s lifetime, Figure 1C. A few time trajectories showing different lifetimes are shown in Appendix A. The interactions between SfiI and DNA duplexes bearing the cognate sites (synaptic specific complexes, *ss*), the DNA duplexes with non-specific sites (non-specific complexes, *nn*), and one duplex with cognate and one without (pre-synaptic complexes, *ns*) were characterized. 

### 2.1. Lifetime of SfiI–DNA Complexes Using 23 bp Substrates

#### 2.1.1. Lifetime of Synaptic Specific Complexes (*ss*)

To characterize the lifetime of synaptic complexes, a DNA duplex with thiol modification bearing the SfiI cognate site was covalently attached to the coverslip, as described in detail in the Methods section. The surface was photobleached and imaged to capture the control data and confirm no fluorescence. Then, SfiI solution was added to the chamber, followed by Cy3-labelled DNA duplex with a cognate site. After the addition of SfiI, the surface was imaged to capture control data and confirm no fluorescence bursts in the absence of the Cy3-labelled DNA substrate. In the control experiment, the Cy3-labelled DNA was added before the SfiI solution, and the fluorescence was recorded to confirm the absence of non-specific interactions between the DNA duplexes or Cy3-DNA and the surface. The formation of a SfiI–DNA synaptosome involving the surface-bound DNA and Cy3-labelled DNA duplexes stabilized by SfiI leads to a fluorescence burst, which is a direct measure of the complex formation. The histogram of 3000 unique fluorescence events in which the data for six independent experiments were combined is shown in Appendix A (please see Methods section for details). The corresponding normalized survival probability with respect to lifetime is shown in Figure 2A, which also contains a single exponential decay fit producing the characteristic lifetime for the synaptic complexes of 29.6 ± 0.4 (sec ± SD). 

#### 2.1.2. Lifetime of Non-Specific Complexes (*nn*)

In these experiments, DNA duplexes with no cognate sites for SfiI were used. The lifetimes of SfiI–DNA complexes with no cognate sites were generated similarly to those described above for synaptic complexes. The histogram of 3000 unique fluorescence events in which the data for six independent experiments were combined is shown in Appendix A. Fitting a single exponential decay function to the normalized survival probability produced 12.6 ± 0.2 (sec ± SD) as the characteristic lifetime for the non-specific complex (Figure 2B).

The SfiI cognate sites on DNA have a high GC content (85%), so to investigate if sequence composition played a role in the complex lifetime, DNA duplexes without a cognate site but with 50% GC content were tethered to the surface and experiments were carried out as described above with a non-specific Cy3-labelled duplex DNA. The set of 1000 fluorescence events for this system collected over three different experiments were recorded and analyzed. Appendix A depicts the histogram of the combined data of all events, which, when fitted with a single exponential decay function, yielded 12.9 ± 0.1 (s) for the characteristic lifetime, Appendix A. This value is indistinguishable from the nonspecific DNA with elevated GC content (Figure 2B), suggesting that GC content is not the factor defining the synaptic complex assembly.

#### 2.1.3. Lifetime of Pre-Synaptic Non-Specific–Specific Complexes (*ns*)

Complexes involving DNA with a cognate site and DNA without a cognate site are essential for the search process. The lifetime of pre-synaptic complexes, consisting of a duplex with a cognate site tethered to the surface and another Cy3-labelled duplex without a cognate site, was characterized using the methodology described above. The histogram of 3000 unique fluorescence events in which the data for six independent experiments were combined is shown in Appendix A. Normalized survival probability fit with a single exponential decay fit yields the characteristic lifetime of non-specific complexes, which is 44.7 ± 0.8 (s), Figure 2C.

The lifetime of pre-synaptic complexes (*ns*) with a flipped surface: To exclude experimental bias from a particular DNA being tethered to the surface, the pre-synaptic experiment was repeated but with the duplex containing the cognate site being Cy3-labelled and the non-specific duplex being tethered to the surface. The histogram of 1000 unique fluorescence events in which the data for three independent experiments were combined is shown in Appendix A. The normalized survival probability with a single exponential decay fit is presented in Appendix A and shows a characteristic lifetime of 42.0 ± 0.6 (s).

The lifetime of pre-synaptic complexes (*ns*) with 50% GC duplex on a surface: To probe if the DNA sequence composition of non-specific duplex affected the lifetime of the presynaptic complexes, we repeated the experiments with a non-specific duplex with 50% GC content tethered to the surface. The histogram of 1000 unique fluorescence events in which the data for three independent experiments were combined is shown in Appendix A. The normalized survival probability fit with a single exponential decay fit (Appendix A), to produce the characteristic lifetime of 44.5 ± 1.5 (s). 

### 2.2. Theoretical Model for SfiI–DNA Complex Lifetimes

In order to explain the unexpected observation of the increased stability of pre-synaptic complexes in comparison with the stability of specific complexes, we developed a phenomenological theoretical approach.

The process of breaking the complex between SfiI and the two DNA duplexes it interacts with is viewed as consisting of two steps: first, the bond with one duplex is broken (the weakest), and this process is reversible, and then the bond with the second duplex is broken. The last irreversible step marks the end of the fluorescence lifetime of the complex, as observed in the experiments. In addition, the following assumptions are made in our theoretical model:(1)After the bond is broken with the first duplex, we assume that the specific/specific (*ss*) complex can only return to the *ss* complex, and no non-specific bonds between protein and DNA molecules can be formed.(2)The non-specific/specific (*ns*) pre-synaptic complex first breaks the bond with the non-specific duplex (the weakest one), and after that, it will have multiple possibilities to return to *ns* complex due to the creation of 23 − 13 + 1 = 11 possible bonds with the non-specific duplex. This also reflects the possibility of sliding in the ns conformation. We call this effect an effective entropic effect. Our hypothesis is that this is the primary source of the larger lifetimes for ns complexes compared to ss complexes. Although a single ns interaction is weaker than the ss interactions, the number of such interactions due to different locations are cumulative, making the overall ns interaction stronger.

The SfiI–DNA complex’s breaking up process can be viewed as an effective two-step chemical-kinetic reaction: state 2 ↔ state 1 → state 0. Here, state i (i = 0, 1, 2) represents the system’s state with i bonds. The complex exists for i = 1, 2, while i = 0 means that the complex is dissociated. 

Let us define *E_n_* and *E_s_* as energies (in units of kT) for making single non-specific and specific bonds between the protein and DNA, respectively. For convenience, we also define the following parameters:(1)$$x=e^{\frac{E_{n}}{2}}\;\;\;y=e^{\frac{E_{s}}{2}}$$

In addition, one can assume that the rate of breaking the non-specific bond is *u*/*x*, and the rate of creating this single bond is *ux*, where *u* is the rate for a chemical process occurring without change in energy. Other distributions of bonding energies between association and dissociation processes are possible, but the specific details will not affect the main conclusions of our theoretical method. It can be shown that:(2)$$\frac{u/x}{ux}=x^{2}=e^{E_{n}}$$
which is the equilibrium constant for the process of making/breaking the bond with the non-specific DNA. 

Similarly, we can write the rates for the specific bond with the DNA molecule.
(3)$$\frac{u/y}{uy}=y^{2}=e^{E_{s}}$$

This is an equilibrium constant for making/breaking the specific bond. 

Now, let us consider the lifetime of the non-specific/non-specific (*nn*) complex. The previous kinetic scheme can describe it, state 2 ↔ state 1 → state 0, with both forward rates given by *u*/*x*, and the single reverse rate given by 11ux. The coefficient 11 comes from the fact that there are 11 different bonds possible with the non-specific DNA where the complex can rebind after the initial bond dissociation. The lifetime can then be evaluated as a mean first-passage time to go from state 0 to state 2, producing:(4)$$T_{nn}=\frac{2x+11x^{3}}{u}$$

We now consider the lifetime of the *ss* complex. In this case, both forward rates are equal to *u*/*y*, and the single backward rate is *uy*. This is because we assumed the system could only return to make two specific bonds again after breaking the first bond. The lifetime, in this case, is given by:(5)$$T_{ss}=\frac{2y+y^{3}}{u}$$

The analysis for the pre-synaptic ns complex produces the following results for the lifetime:(6)$$T_{ns}=\frac{x+y+11x^{2}y}{u}$$

Using the experimental values for *T_nn_*, *T_ss_*, and *T_ns_*, we now have three equations and three variables, *x*, *y*, and *u*, producing the following estimates for the parameters from fitting the experimental data:(7)x≈1.7, y≈5.0,u≈4.6s

These parameters correspond to having the bond energies *E_n_*~0.6 kT and *E_s_*~3.2 kT. 

Thus, our explanation of the unexpectedly large stability of the ns complex is due to a so-called “entropic” (or sliding) effect. There are several sites available for non-specific binding (or sliding) that compensate for the weakness of non-specific binding, making the lifetimes for ns complexes larger than expected. The important advantage of our theoretical method is that it can explicitly calculate the lifetimes of various complexes for different DNA lengths, which can be probed experimentally. For example, for *l* = 33 bp, we predict that:(8)$$T_{nn}\left(theory\right)\approx 20.0s\;,T_{ns}\left(theory\right)\approx 58.0s,\;\;T_{ss}\left(theory\right)\approx 30.1s\;\;$$

Testing of these predictions is described in Section 2.3 below.

In addition, predictions for the mixed-length DNA segments (with *l* = 23 bp and *l* = 33 bp) can be performed. Depending on which DNA segment makes the specific bond, we can theoretically predict that for the mixed systems that:(9)$$\;\;\;\;\;\;\;T_{nn}\left(theory\right)\approx 16.0s\;,T_{ns}\left(theory\right)\approx 58.0s\;or\;36.0s,\;\;T_{ss}\left(theory\right)\approx 30.1s$$

For the *nn* complex, our theory predicts an average between *l* = 23 bp and *l* = 33 bp results. For the ns complex, we predict two different lifetimes that separately correspond to *l*= 23 bp and *l* = 33 bp results. 

The results of experiments designed to test the theoretical predictions are described below. 

### 2.3. Experiment: Lifetime of SfiI–DNA Complexes with 33 bp DNA Substrates

To test the theoretical predictions, we investigated the lifetime of the complexes between SfiI and DNA, which was 33 bp long. The experiments followed the same procedure as with the 23 bp substrates.

#### 2.3.1. SfiI–DNA Complexes with Symmetric Substrates

In this section we have studied the lifetime of SfiI-DNA complexes between Synaptic *(ss)*, Non-Specific *(nn*) and Pre-Synaptic Complexes *(ns)* using the similar methodology as for the 23 bp DNA substrate. Specifically, 33 bp DNA substrates were used in this section and details were provided below.

#### 2.3.2. Lifetime of Synaptic Complexes (*ss*)

TAPIN experiments with SfiI and 33 bp DNA substrates followed the exact same methodology as for the 23 bp DNA substrates. The histogram of 3000 unique fluorescence events in which the data for six independent experiments were combined is shown in Appendix A. The normalized survival probability, fit with a single exponential decay function, is shown in Figure 3A and gives the characteristic lifetime of the 33 bp synaptic complexes with 30.8 ± 0.6 (s). This increase in the lifetime is in line with the theoretical prediction.

#### 2.3.3. Lifetime of the Non-Specific Complexes (*nn*)

The lifetime of non-specific complexes was measured using the same TAPIN approach as for 23 bp substrate. The histogram of 3000 unique fluorescence events in which the data for six independent experiments were combined is shown in Appendix A. A single exponential decay fit of the normalized survival probability of the data produced the characteristic lifetime of 20.1 ± 0.2 (s), Figure 3B.

#### 2.3.4. Lifetime of the Pre-Synaptic Complexes (*ns*)

The lifetime time of the pre-synaptic complexes with 33 bp DNA was then characterized using the described approach. The DNA duplex with no specific site for SfiI was immobilized on the glass surface. The histogram of 3000 unique fluorescence events in which the data for six independent experiments were combined is shown in Appendix A. The normalized survival probability, fit with a single exponential decay, is shown in Figure 3C and yields a characteristic lifetime of 54 ± 0.1 (s).

### 2.4. SfiI–DNA Complexes with Asymmetric Substrates

In this section we have studied the lifetime of SfiI-DNA complexes between Synaptic *(ss)*, Non-Specific *(nn*) and Pre-Synaptic Complexes *(ns)* using the similar methodology as for the 23 bp DNA substrate. Specifically, 23bp and 33 bp DNA substrates were used in this section and details were provided below.

#### 2.4.1. Lifetime of Synaptic Complexes (*ss*)

To characterize the complexes formed between SfiI and two DNA substrates of different lengths, the 23 bp DNA substrate bearing the SfiI specific site was tethered to the glass surface, and the 33 bp duplex with Cy3 and the SfiI cognate site was used as the fluorescent probe. TAPIN experiments were then carried out as described above for synaptic complexes. All experimental steps and procedures were the same as for the symmetric complexes. The histogram of 1000 unique fluorescence events in which the data for three independent experiments were combined is shown in Appendix A. The normalized survival probability of the data is plotted in Figure 4A and gives the characteristic lifetime of 29.8 ± 1 (s) for the asymmetric synaptic complex.

#### 2.4.2. Lifetime of the Non-Specific Complexes (*nn*)

The lifetime of the asymmetric non-specific SfiI–DNA complex was investigated by tethering the 23 bp non-specific duplex to the glass surface and using a Cy3-labelled 33 bp non-specific duplex as the probe. The histogram of 1000 unique fluorescence events in which the data for three independent experiments were combined is shown in Appendix A. The normalized survival probability is plotted in Figure 4B and gives the characteristic lifetime of 14.9 ± 0.3 (s).

#### 2.4.3. Lifetime of the Pre-Synaptic Complexes (*ns*)

Similarly, a 33 bp specific duplex was tethered to the glass surface to investigate the asymmetric pre-synaptic complexes, and the Cy3-labelled 23 bp non-specific duplex was used as the probe. The histogram of 1000 unique fluorescence events in which the data for three independent experiments were combined is shown in Appendix A. The normalized survival probability is shown in Figure 4D and yields the characteristic lifetime of 44 ± 1.2 (s).

We further characterized the reverse complex, with the 23 bp duplex carrying the cognate site being tethered to the glass surface and the non-specific 33 bp duplex being used as the fluorescent probe. The histogram of 1000 unique fluorescence events in which the data for three independent experiments were combined is shown in Appendix A. The normalized survival probability is shown in Figure 4C and gives the characteristic lifetime of 53.6 ± 0.9 (s). 

## 3. Discussion

The results of our studies are summarized in Figure 5 and Table 1. The experiments with DNA duplexes of 23 bp revealed a substantially larger lifetime for specific DNA duplexes (29.6 ± 0.4 s) compared with non-specific ones (12.6 ± 0.2 s), which was also statistically significantly different using the Kolmogorov–Smirnov (KS) test at 0.001 confidence level (*p*-value~10^−133^). According to the theoretical model (Equation (7)), these lifetimes correspond to energies E_s_~3.2 kT and E_n_~0.6 kT, respectively. Notably, according to the theory, due to small binding energy, the complex formed by non-specific duplexes is dynamic, so the protein might frequently change the locations on the duplexes. As a result of such dynamics, the theory predicts that the lifetime of the non-specific complex effectively increases for longer DNA substrate from 16 s to 20 s because of more possibilities for non-specific binding. This prediction perfectly aligns with the experiment produced for the duplexes with 33 bp, the lifetime being 20.1 ± 0.2 s. This observation is the first validation of the theory. Additionally, the approach predicts that the lifetime of a specific complex due to its high stability does not depend on the duplex size, which is also in line with the experiments in which lifetimes for specific duplexes with 23 bp and 33 bp are essentially the same, 29.6 ± 0.4 s and 30.8 ± 0.6 s, respectively.

Moreover, the theory explains the elevated stability of presynaptic complexes, the lifetime of which, 44.7 ± 0.8 s, for the duplexes with 23 bp is considerably higher than the value for the specific complex, 29.6 ± 0.4 s; the KS test shows significant difference between the populations at 0.001 confidence level (*p*-value~10^−25^). According to the theoretical method, the SfiI moves over the non-specific duplex with another binding site of the protein anchored to the cognate site on the specific DNA duplex. Such an entropic effect increases the lifetime for the SfiI pre-synaptic complex with 33 bp duplexes to the value of 54.1 s compared to 30 s for the specific complex, with a significant statistical difference using the KS test at 0.001 confidence level (*p*-value~10^−40^). In the framework of this theoretical model, the presynaptic complex assembled by long specific and short non-specific duplexes should have lifetime values similar to those for the complex formed by short duplexes only. We tested this prediction of the theory by performing experiments with duplexes with 23 bp (non-specific) and 33 bp (specific). The value of 44 ± 1.2 s is essentially the same as the lifetime for duplexes with 23 bp, 44.7 ± 0.8 s. 

Different dynamics of SfiI with specific and non-specific duplexes can be explained by following the structural features of the SfiI synaptosome. SfiI is a homotetramer formed by identical monomers. Each dimer possesses one DNA binding cleft, and the set of dimers binds to two DNA duplexes. According to crystallographic data for SfiI synaptosome [21], the specific DNA duplex in the synaptic complex is bent by up to 25° between the pair of the SfiI dimer. For a non-specific duplex, there is no bend, and the DNA–protein contact is loose, allowing for the duplex to move along another SfiI dimer. 

This model in which SfiI binds dynamically with the non-specific DNA segment explains the search mechanism for SfiI described in our previous work [24]. In this paper, we used time-lapse AFM to visualize the site-search process for SfiI directly. We found that SfiI used a threading pathway in which the enzyme forms a DNA loop and, being bound to one cognate site on DNA, moves (threads) another part of the DNA, changing the loop size. Moreover, the protein stably bound to a specific site on the DNA can dissociate from the non-specific site and jump over to another DNA segment. We termed this translocation pathway as a specific site transfer pathway. With these pathways, the protein translocates over large DNA segments. Previously we observed a threading pathway for the EcoRII restriction enzyme, which also requires binding to two cognate sites to cleave DNA [25,26]. Therefore, we concluded in [24] that other proteins or enzymes could utilize the threading pathway to search for specific sites. This translocation can be in either direction. A recent study has shown a similar observation of increase lifetime with increased length of the DNA and bi-directional translocation (reeling) by the cohesion protein with an increase and decrease of the loop size [31]. Interestingly, DNA loops formed by SfiI and EcoRII during the threading remain relatively flat, so no supercoiling occurs when the protein moves over the distance of serval hundred base pairs that should generate dozens of DNA supercoiling turns [26]. One can expect the supercoiling of DNA rotates when it moves along the protein; such rotation has been proposed for various DNA–protein complexes [32], but we did not find evidence for rotation of SfiI or EcoRII, suggesting that translocation of the protein over DNA can occur without rotation.

## 4. Materials and Methods

### 4.1. Preparation of DNA Duplexes

A total of 9 DNA duplexes with and without the specific site (GGCCTCGAGG GCC) for SfiI were designed with either 23 bp or 33 bp lengths, as shown in Appendix A. The DNA sequences for the duplex were taken from the 3-site DNA substrate used in our previous studies 20–21. Briefly, DNA oligonucleotides, with and without the SfiI specific site and a thiol modification at the 3′ end, were purchased from Integrated DNA Technologies (Coralville, IA, USA) as single-stranded complements. These single-stranded oligonucleotides were stored at −20 °C until use. These single-stranded oligonucleotides were mixed with their complements in a molar ratio of (1:1) and annealed by heating to 98 °C, followed by slow cooling to room temperature. The same approach was used to prepare all the duplexes used in this study for both specific (s) and non-specific (n) duplexes of 23 and 33 bp in length.

The substrate with a cognate site and a thiol modification for surface immobilizing was (5′-TTGGGGGCCTCGAGGGCCATG CC-3′/3ThioMC3-D/); the complementary strand (5′-GGCATG GCCCTCGAGGCCCCCAA-3′) did not contain any modifications. Another substrate with an internal Cy3 fluorophore near the 3′ and a cognate site had the following sequence (5′-TTGGGGGCC TCGAGGGCCAT/iCy3/CC-3′). The same complementary strand as above was used to prepare the duplex.

### 4.2. Annealing and Melting of DNA Duplexes

Melting experiments with the annealed DNA duplexes were performed to confirm the formation of the duplexes. A Varian Cary 50 Bio UV-Visible spectrophotometer (Agilent Technologies, Santa Clara, CA, USA) was used with the following settings: Temperature range (20–95 °C); ramp rate 0.5 deg/min; sampling 1/min; sampling rate 10/point. The melting curve for a 23 bp duplex with a SfiI cognate site and a thiol modification is shown in Appendix A. The melting curve for a duplex with an internally labeled Cy3 is shown in Appendix A. Melting curves for 23 bp duplexes without cognate sites but with thiol or Cy3 modifications are shown in Appendix A, respectively. Similarly, a 23bp duplex with 50% GC with thiol modification but no cognate sites is shown in Appendix A.

Melting experiments for the 33 bp duplexes containing a cognate site with a thiol or Cy3 modification are presented in Appendix A, respectively. For the 33 bp duplexes without cognate sites but with thiol or Cy3 modifications, the melting curves are presented in Appendix A, respectively.

### 4.3. Single-Molecule Fluorescence with TAPIN Approach

#### 4.3.1. Surface Modification

Glass coverslips (180 μm thick, Karl Hecht, Sondheim, Germany) were cleaned in chromic acid for 30 min, followed by multiple rinses with di-water. The coverslips were then treated with 167 μM APS for 30 min, rinsed thoroughly with water, and followed by several rinses with anhydrous DMSO. A freshly prepared mixture (molar ratio of 1:30) of mPEG-SVA (2K):Maleimide-peg-SVA (3.4K) in DMSO was then added to the APS-modified surface and incubated at room temperature for 3 h. The surface was then rinsed with anhydrous DMSO and water. Duplexes with thiol groups were reduced by adding 10 mM TCEP-hydrochloride (Tris(2-carboxyethyl) phosphine) (Pierce, Rockford, IL, USA) to 5 nM DNA in 10 mM TRIS buffer (pH 7.0) and incubating for 30 min at 25 °C. The PEG-functionalized surface was then incubated with the reduced 5 nM duplex DNA solution overnight at 4 °C. The DNA-modified coverslips were thoroughly rinsed and then treated with 10 mM 2-mercaptoethanol in 10 mM HEPES (pH 7.5) for 30 min to quench all non-reacted maleimide groups on the surface, followed by multiple rinses with imaging buffer solution (10 mM HEPES (pH 7.5), 50 mM NaCl, 2 mM CaCl_2_, 0.1 mM EDTA). The modified surface was used immediately after preparation. 

#### 4.3.2. TAPIN Data Acquisition of SfiI–DNA Complexes

The DNA-modified coverslip was assembled in the sample holder (PicoQuant, Berlin, Germany), at the bottom of a sandwich with a 0.1-mm-thick teflon spacer (American Durafilm, Holliston, MA, USA) and 25-mm-diameter quartz disk, with inlet and outlet holes at the top, creating a chamber with 20 mL volume. The assembled sample holder was then mounted on the TIRF microscope.

The chamber was filled with imaging buffer, and the surface was bleached for 30 min. Then, to study the lifetime of SfiI–DNA complexes, SfiI (1 nM) was added to the chamber, followed by a 1nM Cy3-labelled duplex with or without cognate site. Interactions between SfiI and the synaptic complexes (*ss*), the non-specific complexes (*nn*), and pre-synaptic complexes (*ns*) were characterized by matching the duplexes attached to the coverslip with a Cy3-labelled duplex in solution; e.g., to characterize synaptic complexes the coverslip was modified with a duplex containing the cognate site and the Cy3-labelled duplex also contained a cognate site. All TAPIN experiments were performed in triplicates, and each experiment series used two separate surfaces prepared in parallel. 

The following control experiments were performed for each surface: surface alone after bleaching for 30 min (Appendix A); non-specific adsorption of Cy3-labeled DNA onto the DNA-modified surface (Appendix A); and fluorescence caused by the addition of SfiI alone to the DNA-modified surface (Appendix A). In contrast to the formation of the SfiI–DNA complex that shows a significant increase in fluorescent spots (Appendix A), the control experiments did not show any detectable changes. The recorded movies for these control experiments are provided as Appendix A.

TAPIN experiments were performed with an objective-type TIRF microscope built around an Olympus IX71 microscope (Hitschfel Instruments, St. Louis, MO, USA). An oil-immersion UPlanSApo 100× objective with 1.40 NA (Olympus, Tokyo, Japan) was used for all measurements. A laser line at 532 nm (ThorLabs Inc., Newton, NJ, USA) was used to excite the Cy3 labeled DNA. The laser intensity was set to a constant current at 189 mA for all experiments. Fluorescence emission was detected with an electron-multiplying charge-coupled-device camera (ImagEM Enhanced C9100-13, Hamamatsu, Bridgewater, NJ, Westport, CT, USA). TIRF videos of 10-min durations were recorded at a temporal resolution of 100 ms using Micro-Manager v2 software (doi:10.1002/0471142727.mb1420s92).

### 4.4. Data Analysis

TIRF videos were processed and analyzed using ImageJ v1.53t (doi:10.1038/nmeth.2019) with the Template Matching (10.1039/C0LC00641F or https://sites.imagej.net/Template_Matching/, accessed on 5 June 2022) and Spot Intensity analysis (https://github.com/nicost/spotIntensityAnalysis, accessed on 5 June 2022) plugins. Initially, the TIRF movies were subjected to template matching to remove translational drift, followed by spot intensity analysis with threshold XX and pixel size YY to find fluorescence events. A typical SfiI–DNA complex event was detected as a sudden burst in fluorescence intensity followed by an abrupt drop in intensity after a short period of time, as shown in Figure 1C. All detected events with an intensity above 5000 A.U. were selected for analysis, followed by manually sorting the data so that only the intensity data that showed unambiguous fluorescence bursts were included in the final analysis. 

Thousands of events were collected for each SfiI–DNA interaction scenario, and the data was assembled into histograms. The characteristic complex lifetime was then estimated by fitting the histograms with a normalized survival probability analysis [27,28,29,30]. Kolmogorov–Smirnov (KS) test at 0.001 confidence level was performed between different lifetime obtained across different DNA substrates to establish the statistical significant difference.

## 5. Conclusions

The interaction of distant DNA segments is the phenomenon underlying numerous genetic processes accomplished by specialized proteins. A general problem behind accomplishing this work is the site-search process in which the protein brings together two or more distant sites. Pathways proposed for the site-search process of single specific sites on DNA do not explain how the protein brings together distant DNA segments. Our time-lapse AFM studies [20,24,26] revealed two new pathways, which fill this gap and produce a complete picture for the site-search process. However, the molecular mechanism behind these observations remained unclear. In this work, we utilized a single-molecule fluorescence approach to characterize the stability of various complexes corresponding to various transient states of the synaptosome assembly and build the theory of the assembly of various complexes to explain the observations. Our studies revealed a critical role of transient non-specific interaction within the synaptic complexes, which allowed the search process to occur by probing the non-specific DNA segment while another part of the protein is anchored to the specific site. This model explains how the long-range site-search process within the synaptosome occurs, and sheds new light on the role of specific and non-specific interactions within the synaptosome on the site-search mechanism. Our model can be applied to the dynamics of DNA looping mediated by cohesin [33] as dysregulation thereof is implicated in cancer development. This work may serve as a basis for industrial applications in sectors such as biotechnology, pharmaceuticals, and genetic engineering, potentially leading to the development of targeted therapeutics, optimized genetic engineering techniques, enhanced bioproduction processes, and advanced diagnostic technologies.

## Figures and Tables

**Figure 1 ijms-24-09800-f001:**
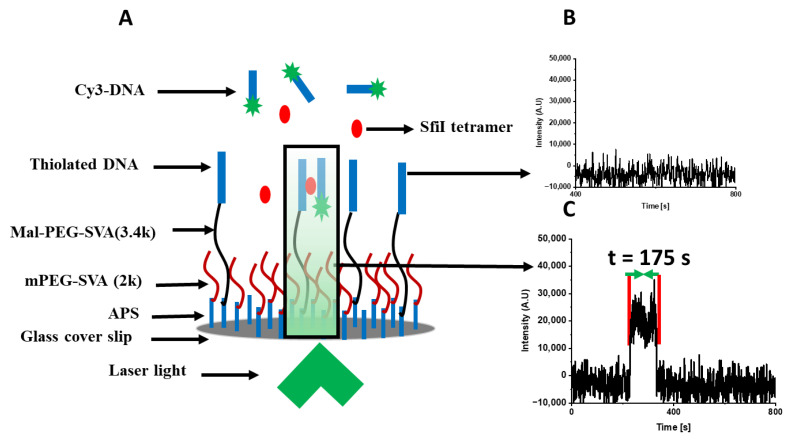
Schematic of TAPIN approach utilized to study the intermolecular interaction of SfiI–DNA using an objective-based TIRF microscope. (**A**) A DNA duplex is tethered to the glass surface through thiol chemistry, SfiI is present in the solution, and a fluorophore-labeled (Cy3) DNA duplex is added to the solution. The DNA substrates cannot interact without SfiI mediating the interaction and will not give any signal (**B**). However, when a complex is formed between SfiI and the two DNA substrates, a fluorescence burst, due to the association and dissociation of a SfiI–DNA complex within the evanescent field, is detected (**C**). The dwell time between association and dissociation events are recorded as the lifetime of the complexes.

**Figure 2 ijms-24-09800-f002:**
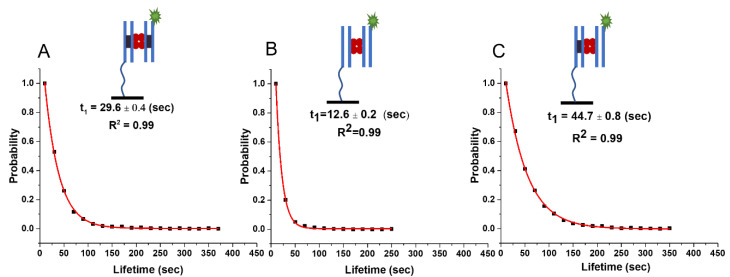
The lifetime of SfiI–DNA complexes with 23 bp DNA substrates. Plots show the normalized survival probability of (**A**) specific–specific, (**B**) non-specific–non-specific, and (**C**) specific–non-specific DNA substrates interacting with SfiI. The characteristic lifetimes of each complex, calculated using a single exponential decay approximation, are given in each plot as a lifetime in sec ± SD. The lifetime of the complexes formed between specific–specific (*ss*) duplexes (synaptic complexes) is 29.6 ± 0.4; between non-specific–non-specific (*nn)* duplexes (non-specific complexes) is 12.6 ± 0.2; and between specific–non-specific (*ns*) duplexes (pre-synaptic complexes) is 44.7 ± 0.8. Two blue lines represent the 23 bp DNA duplex; when the duplex contains a cognate site, it also has a black rectangle inside the blue lines. The fluorescently-labeled DNA substrate has a green star representing the Cy3 label. Four red circles represent the SfiI tetramer. The line drawn from the duplex represents the tether attached to the surface to immobilize the duplex.

**Figure 3 ijms-24-09800-f003:**
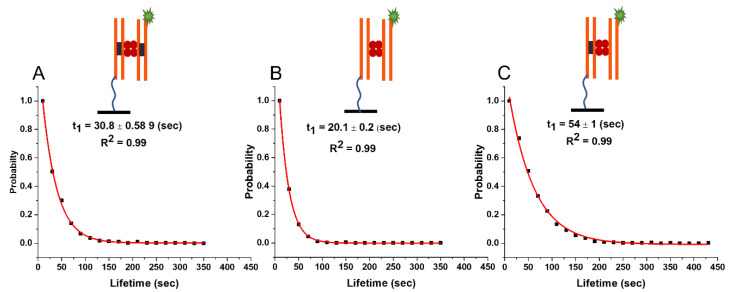
The lifetime of SfiI–DNA complexes with 33 bp DNA substrates. Plots show the normalized survival probability of (**A**) specific–specific, (**B**) non-specific–non-specific, and (**C**) specific–non-specific DNA substrates interacting with SfiI. The characteristic lifetimes of each complex, calculated using a single exponential decay approximation, are given in each plot as a lifetime in sec ± SD. The lifetime of the complexes formed between specific–specific (*ss*) duplexes (synaptic complexes) is 30.8 ± 0.6; between non-specific–non-specific (*nn)* duplexes (non-specific complexes) is 20.1 ± 0.2; and between specific–non-specific (*ns*) duplexes (pre-synaptic complexes) is 54 ± 1. Two orange lines represent the 33 bp DNA duplex; when the duplex contains a cognate site, it also has a black rectangle inside the orange lines. The fluorescently-labeled DNA substrate has a green star representing the Cy3 label. Four red circles represent the SfiI tetramer. The line drawn from the duplex represents the tether attached to the surface to immobilize the duplex.

**Figure 4 ijms-24-09800-f004:**
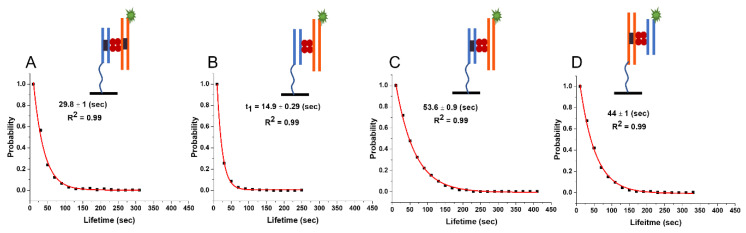
The lifetime of SfiI–DNA complexes with asymmetric DNA substrates of 23 and 33 bp lengths. Plots show the normalized survival probability of (**A**) specific–specific, (**B**) non-specific–non-specific, and (**C**,**D**) specific–non-specific DNA substrates interacting with SfiI. In A–C, the 23 bp substrate is tethered to the surface, while in D, the 33 bp substrate is tethered to the surface. The characteristic lifetime of the complexes formed between DNA duplexes of 23 bp specific (surface) and 33 bp specific (*ss*) (synaptic complexes) is 29.8 ± 1; between the 23 bp non-specific (surface) and 33 bp non-specific (*nn)* duplexes (non-specific complexes) is 20.1 ± 0.2; between 23 bp specific (surface) and 33 bp non-specific (*ns*) duplexes (pre-synaptic complexes) is 53.6 ± 0.9; and between 33 bp specific (surface) and 23 bp non-specific (*ns*) duplexes (pre-synaptic complexes) is 44 ± 1. The line drawn from the duplex represents the tether attached to the surface to immobilize the duplex.

**Figure 5 ijms-24-09800-f005:**
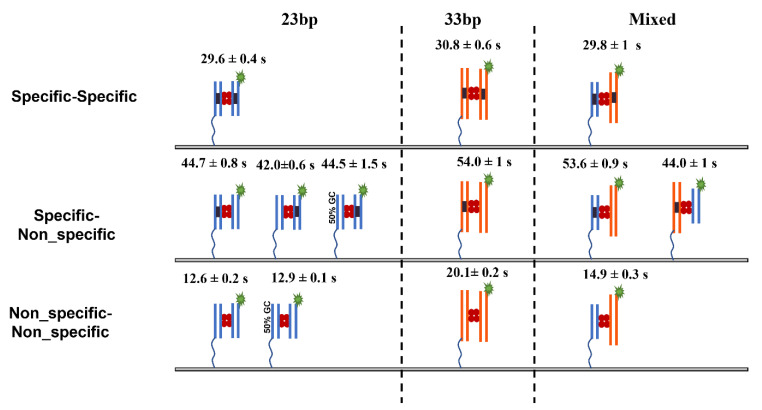
The lifetime of SfiI–DNA complexes. Top row complexes formed between the specific–specific DNA duplexes. In the middle row, complexes formed between the specific and non-specific DNA duplexes, and the complexes formed between 23 bp DNA duplex with 50% GC content. Bottom row complexes formed between the non-specific–non-specific DNA duplexes and the complexes formed between 23 bp DNA duplex with 50% GC content. The line drawn from the duplex represents the tether attached to the surface to immobilize the duplex.

**Table 1 ijms-24-09800-t001:** Summary of the lifetimes for experiments and theory for various types of SfiI–DNA systems.

Complex Type	Experiment 23 bp/23 bp	Experiment 33 bp/33 bp	Experiment 33 bp/23 bp (Fluo)	Theory 33 bp	Theory 23 bp/33 bp
Specific (*ss*)	29.6 ± 0.4 s	30.8 ± 0.6 s	29.8 ± 1 s	30.1 s	30 s
Presynaptic (*ns*)	44.7 ± 0.8 s	54.1 ± 0.1 s	44 ± 1.2 s	58 s	58 s/36 s
Non-specific (*nn*)	12.6 ± 0.2 s	20.1 ± 0.2 s	14.9 ± 0.3 s	20 s	16 s

## Data Availability

The authors declare that the data supporting the findings of this study are available within the paper and its Appendix A files.

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
