# Peer review of "Assembly of Synaptic Protein–DNA Complexes: Critical Role of Non-Specific Interactions"

_ijms, 2023, doi:10.3390/ijms24129800_

Round 1

Reviewer 1 Report

The authors have evaluated the stability of various complexes involved in synaptosome formation using the single molecule fluorescence method, using signal duration as an indicator. As a result, they predicted and constructed the various complexes involved in stability. These studies have revealed the importance of transient, nonspecific interactions within synaptic complexes. This suggests that one experimental evidence can be obtained for a process that searches for sites of protein complex formation. I find this method particularly interesting as it provides a new research perspective on the role of specific and nonspecific interactions between proteins and DNA in the search process between long-distance sites, i.e., distant sites on DNA. The experimental methods and the discussion of the experimental results are also scientifically well reviewed, and I strongly recommend that the paper be published in this journal.

No modifications to the English are necessary since the manuscript is sufficiently comprehensible.

Author Response

We thank the reviewer for positive comments on the paper.

Reviewer 2 Report

The submitted work studies the lifetime associated to DNA-SfiI complexes by total internal reflection fluorescence (TIRF). This work is built in previous work recently published by the authors (doi: 10.3390/ijms23010212). The authors should highlight the originality of this work since there exist many other examples in literature where employ the same approach used by the authors [1] or even more sophisticated methods [2]:

[1] Schaich, M.A.; et al. Single molecule analysis of DNA-binding proteins from nuclear extracts (SMADNE). Nucleic Acids Res. 2023, 51, e39. https://doi.org/10.1093/nar/gkad095.

[2] Böger, C.; et al. Super-resolution imaging and estimation of protein copy numbers at single synapses with DNA-point accumulation for imaging in nanoscale topography. Neurophotonics 2019, 6, 035008. https://doi.org/10.1117/1.NPh.6.3.035008.

However, it exists some points that need to be addressed (please, see them below detailed point-by-point). Here, there exists some suggestions in order to improve the scientific quality of the manuscript paper:

MAJOR REVISIONS

1) The authors must discussed the originality of this work regarding the research previously devoted in other published manuscripts. The Introduction section is too short and this point may significantly aid to gain the attention of the potential readers.

2) “Thousands of events were collected for each SfiI-DNA interaction scenario, and the data was assembled into histograms. The characteristic complex lifetime was then estimated by fitting the histograms with a normalized survival probability analysis” (lines 46-49). These data appear as SI. If I understand well, the authors only records data for further processing of one sample measurement. Why did they not replicate the measurement several times (e.g. N = 3) in order to conduct more robust statistical data analysis?

3) Table 1 (line 317). The authors should devote Student’s t-test or analysis of variance (ANOVA) to discern if the observed differences between the tested conditions are significally or not.

MINOR REVISIONS

4) INTRODUCTION. The authors should mention other DNA-protein systems like the examples of regulatory factors [3], nucleosome formation [4] or the degradosome assembly [5] where the used approach could be used to reveal the respective complex lifetimes.

[3] Pallarés, M.C.; et al. Sequential binding of FurA from Anabaena sp. PCC 7120 to iron boxes: exploring regulation at the nanoscale. Biochim. Biophys. Acta. 2014, 1844, 623-631. https://doi.org/10.1016/j.bbapap.2014.01.005.

[4] Makowski, M.M.; et al. Global profiling of protein-DNA and protein-nucleosome binding affinities using quantitative mass spectrometry. Nat. Commun. 2018, 9, 1653. https://doi.oirg/10.1038/s41467-018-04084-0.

[5] Novo, N.; et al. Beyond a platform protein for the degradosome assembly: The Apoptosis-Inducing Factor as an efficient nuclease involved in chromatinolysis. PNAS Nexus 2022, 2, pgac312. https://doi.org/10.1093/pnasnexus7pgac312.

5) RESULTS. Figure 1 (line 91). The figure is slighty blurry maybe caused during the pdf conversion step. The point should be taken into account not only by the authors but also the layout staff of the journal.

6) “The surface was photobleached for 30 mins and imaged to capture the control data and confirm no fluorescence. Then, 1 nM SfiI solution was added to the chamber, followed by 1nM Cy3-labelled DNA duplex with a cognate site” (lines 104-106). It would be more appropiate to place this information in the Materials & Methods section.

7) (lines 229-237). Where is the Equation number 9? Then, “(…) [10]c” (line 237). Please, the authors should erase the letter c appeared after the brackets.

8) “(…) yields a characteristic lifetime of 54 ± 0.1 (sec) (line 265). Please, the authors should homogenize the significat figures of the displayed data. Same comment for the Table 1 (line 317) and the rest of the manuscript body text.

9) CONCLUSIONS. How the society could be benefited of this conducted work? What are some potential future avenues to pursue this research? Will the most relevant outcomes of this work serve as a basis for any Industrial application? The authors should briefly discuss about these points and provide their point of view.

10) REFERENCES. The references are not in the proper format style of International Journal of Molecular Sciences. The names of some journals should appear in abbreviated form. Furthermore, the publication issue should be erased.

The English is fine.

Round 2

Reviewer 2 Report

The scientific quality of the revised manuscript version has significantly improved. Nevertheless, I still find issues about the originality of this manuscript. Based on the high-standards of International Journal of Molecular Sciences, I warmly recommend a transfer desk option.

The English is fine.